# TimbreTron: A WaveNet(CycleGAN(CQT(Audio))) Pipeline for Musical Timbre Transfer

**Sicong Huang[1,2], Qiyang Li[1,2], Cem Anil[1,2], Xuchan Bao[1,2], Sageev Oore[2,3], Roger B. Grosse[1,2]**
University of Toronto[1], Vector Institute[2], Dalhousie University[3]

## Abstract

In this work, we address the problem of musical timbre transfer, where the goal is to manipulate the timbre of a sound sample from one instrument to match another instrument while preserving other musical content, such as pitch, rhythm, and loudness. In principle, one could apply image-based style transfer techniques to a time-frequency representation of an audio signal, but this depends on having a representation that allows independent manipulation of timbre as well as high-quality waveform generation. We introduce TimbreTron, a method for musical timbre transfer which applies "image" domain style transfer to a time-frequency representation of the audio signal, and then produces a high-quality waveform using a conditional WaveNet synthesizer. We show that the Constant Q Transform (CQT) representation is particularly well-suited to convolutional architectures due to its approximate pitch equivariance. Based on human perceptual evaluations, we confirmed that TimbreTron recognizably transferred the timbre while otherwise preserving the musical content, for both monophonic and polyphonic samples. We made an accompanying demo video [1] which we strongly encourage you to watch before reading the paper.

## 1 Introduction

Timbre is a perceptual characteristic that distinguishes one musical instrument from another playing the same note with the same intensity and duration. Modeling timbre is very hard, and it has been referred to as "the psychoacoustician's multidimensional waste-basket category for everything that cannot be labeled pitch or loudness"[2]. The timbre of a single note at a single pitch has a nonlinear dependence on the volume, time and even the particular way the instrument is played by the performer. While there is a substantial body of research in timbre modelling and synthesis (Chowning (1973); Risset and Wessel (1999); Smith (2010; 2011)), state-of-the-art musical sound libraries used by orchestral composers for analog instruments (e.g. the Vienna Symphonic Library (GmbH, 2018)) are still obtained by extremely careful audio sampling of real instrument recordings. Being able to model and manipulate timbre electronically carries importance for musicians who wish to experiment with different sounds, or compose for multiple instruments. (Appendix A discusses the components of music in more detail.)

In this paper, we consider the problem of high quality *timbre transfer* between audio clips obtained with different instruments. More specifically, the goal is to transform the timbre of a musical recording to match a set of reference recordings while preserving other musical content, such as pitch and loudness. We take inspiration from recent successes in style transfer for images using neural networks (Gatys et al., 2015; Johnson et al., 2016; Ulyanov et al., 2016; Chu et al., 2017). An appealing strategy would be to directly apply image-based style transfer techniques to time-frequency representations of images, such as short-time Fourier transform (STFT) spectrograms. However, needing to convert the generated spectrogram into a waveform presents a fundamental obstacle, since accurate reconstruction requires phase information, which is difficult to predict (Engel et al., 2017), and existing techniques for inferring phase (e.g., Griffin and Lim (1984)) can produce characteristic artifacts which are undesirable for high quality audio generation (Shen et al., 2017).

---

[1]Link to the demo video: www.cs.toronto.edu/~huang/TimbreTron/index.html
  Code available at: https://github.com/huangsicong/TimbreTron
[2]McAdams and Bregman (1979), pg 34

Recent years have seen rapid progress on audio generation methods that directly generate high-quality waveforms, such as WaveNet (van den Oord et al., 2016), SampleRNN (Mehri et al., 2016), and Tacotron2 (Shen et al., 2017). WaveNet's ability to condition on abstract audio representations is particularly relevant, since it enables one to perform manipulations in high-level auditory representations from which reconstruction would have previously been impractical. Tacotron2 performs high-level processing on time-frequency representations of speech, and then uses WaveNet to output high-quality audio conditioned on the generated mel spectrogram.

We adapt this general strategy to the music domain. We propose TimbreTron, a pipeline that performs CQT-based timbre transfer with high-quality waveform output. It is trained only on unrelated samples of two instruments. For our time-frequency representation, we choose the constant Q transform (CQT), a perceptually motivated representation of music (Brown, 1991). We show that this representation is particularly well-suited to musical timbre transfer and other manipulations due to its pitch equivariance and the way it simultaneously achieves high frequency resolution at low frequencies and high temporal resolution at high frequencies, a property that STFT lacks.

TimbreTron performs timbre transfer by three steps, shown in Figure 1. First, it computes the CQT spectrogram and treats its log-magnitude values as an image (discarding phase information). Second, it performs timbre transfer in the log-CQT domain using a CycleGAN (Zhu et al., 2017). Finally, it converts the generated log-CQT to a waveform using a conditional WaveNet synthesizer (which implicitly must infer the missing phase information). Empirically, our TimbreTron can successfully perform musical timbre transfer on some instrument pairs. The generated audio samples have realistic timbre that matches the target timbre while otherwise expressing the same musical content (e.g., rhythm, loudness, pitch). We empirically verified that the use of a CQT representation is a crucial component in TimbreTron as it consistently yields qualitatively better timbre transfer than its STFT counterpart.

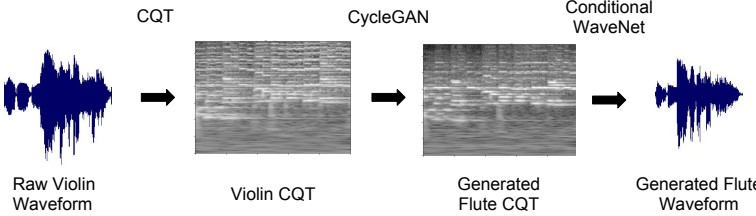

Figure 1: The TimbreTron pipeline that performs timbre transfer from Violin to Flute.

## 2 BACKGROUND

### 2.1 TIME-FREQUENCY ANALYSIS

Time-frequency analysis refers to techniques that aim to measure how the signal's frequency domain representation changes over time.

**Short Time Fourier Transform (STFT)** The STFT is one of the most commonly applied techniques for this purpose. The discrete STFT operation can be compactly expressed as follows:

$$STFT\{x[n]\}(m, \omega_k) = \sum_{n=-\infty}^{\infty} x[n]w[n-m]e^{-j\omega_k n}$$

The above formula computes the STFT of an input time-domain signal $x[n]$ at time step $m$ and frequency $\omega_k$. $w$ refers to a zero-centered window function (such as Hann Window), which acts as a means of masking out the values that are away from $m$. Hence, the equation above can be interpreted as the discrete Fourier transform of the masked signal $x[n]w[n-m]$. An example spectrogram is shown in Figure 2.

**Constant Q Transform (CQT).** The CQT (Brown, 1991) is another time-frequency analysis technique in which the frequency values are geometrically spaced, with the following particular pat-

tern (Blankertz): $\omega_k = 2^{\frac{k}{b}} \omega_0$. Here, $k \in \{1, 2, 3, ...k_{max}\}$ and $b$ is a constant that determines the geometric separation between the different frequency bands. To make the filter for different frequencies adjacent to each other, the bandwidth of the $k^{th}$ filter is chosen as: $\Delta_k = \omega_{k+1} - \omega_k = \omega_k(2^{\frac{1}{b}} - 1)$. This results in a constant frequency to resolution ratio (as known as the "quality (Q) factor"):

$$Q = \frac{\omega_k}{\Delta_k} = (2^{\frac{1}{b}} - 1)^{-1}$$

Huzaifah (2017) showed that CQT consistently outperformed traditional representations such as Mel-frequency cepstral coefficients (MFCCs) in environmental sound classification tasks using CNNs.

**Rainbowgram.** Engel et al. (2017) introduced the rainbowgram, a visualization of the CQT which uses color to encode time derivatives of phase; this highlights subtle timbral features which are invisible in a magnitude CQT. Examples of CQTs and rainbowgrams are shown in Figure 2.

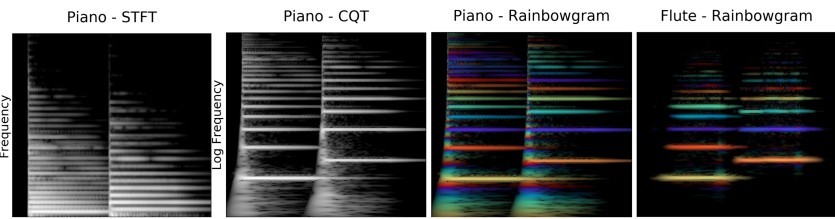

Figure 2: The STFT of a piano clip **(left)**, the CQT of the same piano clip **(second left)**, the rainbowgram of the same piano clip **(second right)** and the rainbowgram of a flute clip which has the same pitch as the first piano clip **(right)**. Note that the harmonics of different pitches are **approximate translations** of each other in the CQT representation.

## 2.2 WAVEFORM RECONSTRUCTION FROM SPECTROGRAMS

Synthesis (waveform reconstruction) from the aforementioned time-frequency analysis techniques can be performed in the presence of both magnitude and phase information (Allen and Rabiner, 1977) (Holighaus et al., 2013). In the absence of phase information, one of the common methods of synthetically generating phase from STFT magnitude is the Griffin-Lim algorithm (Griffin and Lim, 1984). This algorithm works by randomly guessing the phase values, and iteratively refining them by performing STFT and inverse STFT operations until convergence, while keeping the magnitude values constant throughout the process. Developed to minimize the mean squared error between the target spectrogram and predicted spectrogram, this algorithm is shown to reduce the objective function at each iteration, while having no optimality guarantees due to the non-convexity of the optimization problem (Griffin and Lim, 1984; Sturmel and Daudet) Although recent developments in the field have enabled performing the inverse operation of CQT (Velasco et al., 2011; Fitzgerald et al., 2006), these techniques still require both phase and magnitude information.

## 2.3 WAVENET

WaveNet, proposed by van den Oord et al. (2016), is an auto-regressive generative model for generating raw audio waveform with high quality. The model consists of stacks of dilated causal convolution layers with residual and skip connections. WaveNet can be easily modified to perform conditional waveform generation; for example, it can be trained as a vocoder for synthesizing natural, high-quality human speech in TTS systems from low-level acoustic features (e.g., phoneme, fundamental frequency, and spectrogram) (Arik et al., 2017; Shen et al., 2017). One limitation of WaveNet is that the generation of waveforms can be expensive, which is undesirable for training procedures that require auto-regressive generation (e.g., GAN training, scheduled sampling).

## 2.4 GAN AND CYCLEGAN

Generative Adversarial Networks (GANs) are a class of implicit generative models introduced by Goodfellow et al. (2014). A GAN consists of a discriminator and a generator, which are trained

adversarially via a two-player min-max game, where the discriminator attempts to distinguish real data from samples, and the generator attempts to fool the discriminator. The objective is:

$$G^*, D^* = \arg\min_G \max_D \mathbb{E}_{x \sim \mathcal{X}}[\log D(x)] + \mathbb{E}_{z \sim \mathcal{Z}}[\log(1 - D(G(z)))], \tag{1}$$

where $D$ is the discriminator, $G$ is the generator, $z$ is the latent code vector sampled from Gaussian distribution $\mathcal{Z}$, and $x$ is sampled from data distribution $\mathcal{X}$. GANs constituted a significant advance over previous generative models in terms of the quality of the generated samples.

CycleGAN (Zhu et al., 2017) is an architecture for unsupervised domain transfer: learning a mapping between two domains without any paired data. (Similar architectures were proposed independently by Yi et al. (2017); Liu et al. (2017); Kim et al. (2017).) The CycleGAN learns two generator mappings: $F : \mathcal{X} \to \mathcal{Y}$ and $G : \mathcal{Y} \to \mathcal{X}$; and two discriminators: $D_{\mathcal{X}} : \mathcal{X} \to [0, 1]$ and $D_{\mathcal{Y}} : \mathcal{Y} \to [0, 1]$. The loss function of CycleGAN consists of both adversarial losses (Eqn. 1), combined with a cycle consistency constraint which forces it to preserve the structure of the input:

$$\mathcal{L}_{\text{cyc}}(F, G, \mathcal{X}, \mathcal{Y}) = \mathbb{E}_{x \sim \mathcal{X}}[\|G(F(x)) - x\|_1] + \mathbb{E}_{y \sim \mathcal{Y}}[\|F(G(y)) - y\|_1] \tag{2}$$

## 3 MUSIC PROCESSING WITH CONSTANT-Q-TRANSFORM REPRESENTATION

This section focuses on the first and last steps of the TimbreTron pipeline: the steps related to the transforming raw waveforms to and from time frequency representations. We explain our reasoning for choosing the CQT representation and introduce our conditional WaveNet synthesizer which converts a (possibly generated) CQT to a high-quality audio waveform.

### 3.1 CQT FOR MUSIC REPRESENTATION

The CQT representation (Brown, 1991) has desirable characteristics that make it especially suitable for processing musical audio signals. It uses a logarithmic representation of frequency, where the frequencies are generally chosen to exactly cover all the pitches present in the twelve tone, well-tempered scale. Unlike the STFT, the CQT has higher frequency resolution towards lower frequencies, which leads to better pitch resolution for lower register instruments (such as cello or trombone), and higher time resolution towards higher frequencies, which is advantageous for recovering the fine timing of rhythms. Since individual notes contain information across many frequencies (due to their pattern of overtones), this combination of resolutions ought to allow simultaneous recovery of pitch and timing information for any particular note. (While this information is preserved in the signal, waveform recovery is a difficult problem in practice; this is discussed in Section 3.2).

Another key feature of the CQT representation in the context of TimbreTron is (approximate) pitch equivariance. Thanks to the geometric spacing of frequencies, a pitch shift corresponds (approximately) to a vertical translation of the "spectral signature" (unique pattern of harmonics) of musical instruments. This means that the convolution operation is approximately equivariant under pitch translation, which allows convolutional architectures to share structure between different pitches. A demonstration of this can be seen in Figure 3. Since the harmonics of a musical instrument are approximately integer multiples of the fundamental frequency, scaling the fundamental frequency (hence the pitch) corresponds to a constant shift in all of the harmonics in log scale.

We also want to emphasize on some of the reasons why the equivariance is only approximate:

- **Imperfect multiples:** In real audio samples from instruments, the harmonics are only approximately integer multiples of the fundamental frequency, due to the material properties of the instruments producing the sound.

- **Dependence of spectral signature on pitch and beyond:** For each pitch, each instrument has a slightly different spectral signature, meaning that a simple translation in the frequency axis cannot completely account for the changes in the frequency spectrum. Furthermore, even at a given pitch it can still change depending on how it's played.

We used 16ms frame hop (256 time steps under 16kHz). More details can be found in Appendix B.

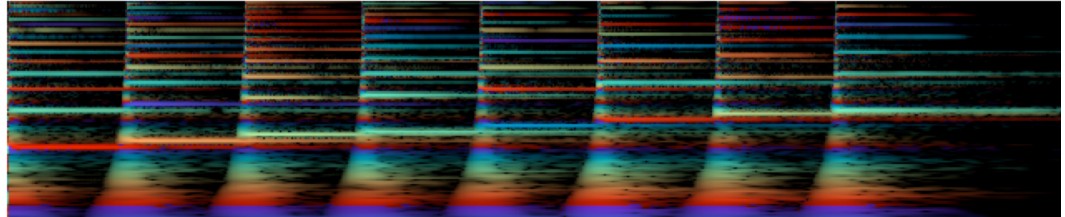

Figure 3: The rainbowgram of a C major scale played by piano.

## 3.2 Waveform Reconstruction from CQT Representation using Conditional WaveNet

Since empirical studies have shown it is difficult to directly predict phase in time-frequency representations (Engel et al., 2017), we discard the phase information and perform the image-based processing directly on a log-amplitude CQT representation. Therefore, in order to recover a waveform consistent with the generated CQT, we need to infer the missing phase information, which is a difficult problem (Velasco et al., 2011).

To convert log magnitude CQT spectrograms back to waveforms, we use a 40-layer conditional WaveNet with the dilation rate of $2^{k \pmod{10}}$ for the $k^{th}$ layer. The model is trained using pairs of a CQT and a waveform; this requires only a collection of unlabeled waveforms, since the CQT can be computed from the waveform.[3] See Appendix C.4 for the details of the WaveNet architecture. WaveNet reconstructed audio samples can be found here[4]

**Beam Search** Because the conditional WaveNet generates stochastically from its predictive distribution, it sometimes produces low-probability outputs, such as hallucinated notes. Also, because it has difficulty modeling the local loudness, the loudness often drifts significantly over the timescale of seconds. While these issues could potentially be addressed by improving the WaveNet architecture or training method, we instead take the perspective that the WaveNet's role is to produce a waveform which matches the target CQT. Since the above artifacts are macro-scale errors which happen only stochastically, the WaveNet has a significant probability of producing high-quality outputs over a short segment (e.g. hundreds of milliseconds). Therefore, we perform a beam search using the WaveNet's generations in order to better match the target CQT. See Appendix C.5 for more details about our beam search procedure.

**Reverse Generation** In early experiments, we observed that percussive attacks (onset characteristics of an instrument in which it reaches a large amplitude quickly) are sometimes hard to model during forward generation, resulting in multiple attacks or missing attacks. We believe this problem occurs because it is difficult to determine the onset of a note from a CQT spectrogram (in which information is blurred in frequency), and it is difficult to predict precise pitch at the note onset due to the broad frequency spectrum at that moment. We found that the problems of missing and doubled attacks could be mostly solved by having the WaveNet generate the waveform samples in reverse order, from end to beginning.

## 4 Timbre Transfer with CycleGAN on CQT Representation

In this section, we describe the middle step of our TimbreTron pipeline, which performs timbre transfer on log-amplitude CQT representations of the waveforms. As training data, we have collections of unrelated recordings of different musical instruments. Hence, our timbre transfer problem on log-amplitude CQT "images" is an instance of unsupervised "image-to-image" translation. To achieve

---

[3]We up-sample the CQT spectrograms to the rate of the audio using nearest neighbour interpolation before conditioning them to the WaveNet. The audio sample is quantized using 8-bit mu-law, and the output of the WaveNet is from softmax layer over 256 quantized values. At test time, we run the conditional WaveNet autoregressively with the initial condition of zero as the first sample value.

[4]Link to WaveNet reconstructed samples www.cs.toronto.edu/~huang/TimbreTron/others.html#wavenet

this, we applied the CycleGAN architecture, but adapted it in several ways to make it more effective for time-frequency representations of audio.

**Removing Checkerboard Artifacts**   The convnet-resnet-deconvnet based generators from the original CycleGAN led to significant checkerboard artifacts in the generated CQT, which corresponds to severe noise in the generated waveform. To alleviate this problem, we replaced the deconvolution operation with nearest neighbor interpolation followed with regular convolution, as recommended by Odena et al. (2016).

**Full-Spectrogram Discriminator**   Due to the local nature of the original CycleGAN's transformations, Zhu et al. (2017) found it advantageous for the discriminator only to process a local patch of the image. However, when generating spectrograms, it's crucial that different partials of the same pitch be consistent with each other; a discriminator which is local in frequency cannot enforce this. Therefore, we gave the discriminator the full spectrogram as input.

**Gradient Penalty(GP)**   Replacing the patch discriminator with the full-spectrogram one led to unstable training dynamics because the discriminator was too powerful. To compensate for this, we added the Gradient Penalty(GP) (Gulrajani et al., 2017) to enforce a soft Lipschitz constraint:

$$\mathcal{L}_{\text{GP}}(G, D, \mathcal{Z}, \hat{\mathcal{X}}) = \alpha \cdot \mathbb{E}_{\hat{x} \sim \hat{\mathcal{X}}}[(\|\nabla_{\hat{x}} D(\hat{x})\|_2 - 1)^2] \tag{3}$$

Here $\hat{\mathcal{X}}$ are samples taken along a line between the true data distribution $\mathcal{X}$ and the generator's data distribution $\mathcal{X}_g = \{F(z)|z \sim \mathcal{Z}\}$ via convex combination of a real data point and a generated data point. Fedus et al. (2018) showed empirically that the GP can stabilize GAN training. Furthermore, Gomez et al. (2018) showed that GP can also stabilize and improve CycleGAN training with word embeddings. We observed the same benefits in our experiments.

**Identity loss**   In addition to the adversarial loss and the reconstruction loss that we applied to the generators, we also added identity loss, which was proposed by Zhu et al. (2017) to preserve color composition in the original CycleGAN. Empirically, we found out that the identity loss component helps generators to preserve music content, which yields better audio quality empirically.

$$\mathcal{L}_{\text{identity}}(F, G, \mathcal{X}, \mathcal{Y}) = \mathbb{E}_{x \sim \mathcal{X}}[\|F(x) - y\|_1] + \mathbb{E}_{y \sim \mathcal{Y}}[\|G(y) - x\|_1] \tag{4}$$

Our weighting of the identity loss followed a linear decay schedule (details in Appendix C.3). In this way, at the start of training, the generator is encouraged to learn a mapping that preserves pitch; as training progresses, the enforcement is reduced, allowing the generator to learn more expressive mappings.

See Appendix C.2, C.3, and C.6 for more details of our CycleGAN architecture, and training and generation methods.

## 5   RELATED WORK

There is a long history of using clever representations of images or audio signals in order to perform manipulations which are not straightforward on the raw signals. In a seminal work, Tenenbaum and Freeman (1999) used a multilinear representation to separate style and content of images. Ulyanov and Lebedev (2016) and Verma and Smith (2018) then applied the optimization technique proposed by Gatys et al. (2015) to the audio domain by applying the image-based architectures to spectrogram representations of the signals. Grinstein et al. (2017) took a similar approach, but used hand-crafted features to extract statistics from the spectrograms. However, a recent review by Dai et al. (2018) pointed out that the disentanglement of timbre and performance control information remains unsolved.

Zhu et al. (2017) introduced Cycle GAN approach to learn an "unsupervised image-to-image mapping" between two unpaired datasets using two generator networks and two discriminator networks with generative adversarial training. Given the success of the CycleGAN on image domain style transfer, Kaneko and Kameoka (2017) applied the same architecture to translate between human voices in the Mel-cepstral coefficient (MCEP) domain and Brunner et al. (2018) applied it to musical style transfer with MIDI representations.

What the aforementioned audio style transfer approaches have in common is that the reconstruction quality is limited by the existing non-parametric algorithms for audio reconstruction (e.g., the Griffin-Lim algorithm for STFT domain reconstruction (Griffin and Lim, 1984), or the WORLD vocoder for MCEP domain reconstruction of speech signals (Morise et al., 2016)), or existing MIDI synthesizer.

Another strategy is to operate directly on waveforms. van den Oord et al. (2016) demonstrated high-quality audio generation using WaveNet. Following on this, Engel et al. (2017) proposed a WaveNet-style autoencoder model operating on raw waveforms that was capable of creating new, realistic timbres by interpolating between already existing ones. Donahue et al. (2018) proposed a method to synthesize waveforms directly using GANs with improved quality over naive generative models such as SampleRNN (Mehri et al., 2016) and WaveNet. Mor et al. (2018) used an encoder-decoder approach for the Timbre Transfer problem, where they trained a universal encoder to learn a shared representation of raw waveforms of various instruments, as well as instrument-specific decoders to reconstruct waveforms from the shared representation. In a parallel work, Bitton et al. (2018) approached the many-to-many timbre transfer problem with their MoVE model which is based on UNIT (Liu et al., 2017) but with Maximum Mean Discrepancy (MMD) as their objective. While their approach has the advantage of training a single model for many transfer directions, our TimbreTron model has the advantage that it uses a GAN-based training objective, which (in the image domain) typically results in outputs with higher perceptual quality compared to VAEs.

## 6 EXPERIMENTS

We conducted two sets of experiments to 1) experiment with pitch-shifting and tempo-changing to further validate our choice of CQT representation; 2) test our full TimbreTron pipeline (along with ablation experiments to validate our architectural choices). See Appendix C for the details of our experimental setup. For this section, please listen to audio samples we provided in our website[5] as you read along.

### 6.1 DATASETS

Training TimbreTron requires collections of unrelated recordings of the source and target instruments. We built our own MIDI and real world datasets of classical music for training TimbreTron. Within each type of dataset, we gathered unrelated recordings of Piano, Flute, Violin and Harpsichord and then divided the entire dataset into training set and test set. We ensured that the training and test sets were entirely disjoint in terms of musical content by splitting the datasets by musical piece. The training dataset was divided into 4-second chunks, which were the basic units processed by our CycleGAN and WaveNet. Links to source audio and more details about our dataset are given in Appendix C.1

### 6.2 DISENTANGLING PITCH AND TEMPO USING CQT REPRESENTATION

Before presenting our timbre transfer results, we first consider the simpler task of disentangling pitch and tempo. Recall that the two properties are entangled in the time domain representation, e.g. subsampling the waveform simultaneously increases the tempo and raises the pitch. Changing the two independently requires more sophisticated analysis of the signal. In the context of our TimbreTron pipeline, due to the CQT's pitch equivariance property, **pitch shifting** can be (approximately) performed simply by translating the CQT representation on the log-frequency axis. (Since the STFT uses linearly sampled frequencies, it does not lend itself easily to this type of simple transformation.) **Audio time stretching** can be done using either the CQT or STFT representations, combined with the WaveNet synthesizer, by changing the number of waveform samples generated per CQT window. Regardless of the number of samples generated, the WaveNet synthesizer is able to produce the correct pitch based on the local frequency content. (See section 6.2 of the OneDrive folder) In conclusion, our method was able to vary the pitch and tempo independently while otherwise preserving the timbre and musical structure.

---

[5]Link to the final samples: `https://www.cs.toronto.edu/~huang/TimbreTron/samples_page.html`

### 6.3 Timbre Transfer Experiments

While most of our experiments on timbre transfer are conducted on real world music recordings, we also use synthetic MIDI audio data in our ablation studies because it is possible to produce paired dataset for evaluation purpose. In this section, we show our experimental findings on the full TimbreTron pipeline using real world data, verify the correctness of our reasoning about CQT, and show the generalization capability of TimbreTron.

**Comparing CQT and STFT Representations**   One of the key design choices in TimbreTron was whether to use an STFT or CQT representation. If the STFT representation is used, there is an additional choice of whether to reconstruct using the Griffin-Lim algorithm or the conditional WaveNet synthesizer. We found that the STFT-based pipeline had two problems: 1) it sometimes failed to correctly transfer low pitches, likely due to the STFT's poor frequency resolution at low frequencies, and 2) it sometimes produced a random permutation of pitches. For example, we ran TimbreTron on a Bach piano sample played by a professional musician. The STFT TimbreTron transposed parts of the longer excerpt by different amounts, and for a few notes in particular, seemed to fail to transpose them by the same amount as it did the others. As is shown by audio samples here[6], those problems were completely solved using CQT TimbreTron (likely due to the CQT's pitch equivariance and higher frequency resolution at low frequencies). Both of these artifacts occurred in both WaveNet and Griffin-Lim reconstruction methods (See Table 4), which suggests that the source of the artifacts are likely to be from the CycleGAN stage of the pipeline. (Please listen to corresponding samples in section 6.3 of the OneDrive folder) This empirically demonstrates the effectiveness of the CQT representation compared with STFT.

**Generalizing from MIDI to Real-World Audio**   To further explore the generalization capability of TimbreTron, we also tried one domain adaptation experiment where we took a CycleGAN trained on MIDI data, tested it on the real world test dataset, and synthesized audio with Wavenet trained on training real world data. As is shown from the corresponding audio examples in this section[7], the quality of generated audio is very good, with pitch preserved and timbre transfered. The ability to generalize from MIDI to real-world is interesting, in that it opens up the possibility of training on paired examples.

### 6.4 Evaluation with Amazon Mechanical Turk (AMT)

We conducted a human study to investigate whether TimbreTron could transfer the timbre of a reference collection of signals while otherwise preserving the musical content. We also evaluated the effectiveness of the CQT representation by comparing with a variant of TimbreTron with the CQT replaced by the STFT. All results are shown in Tables 2, 3 and 4, with detailed discussion in this section. A list of questions asked in AMT can be found in Table 1.

**Does TimbreTron transfer timbre while preserving the musical piece?**   To be effective, the system must transform a given audio input so that the output is (1) recognizable as the same (or appropriately similar) basic musical piece, and (2) recognizable as the target instrument. We address both of these criteria by two types of comparison-based experiments: instrument similarity and musical piece similarity. The questions we asked are listed in Table 1. Table 2 shows results for the instrument similarity comparison and Table 3 shows results for the music piece similarity comparison. The respondents were also asked to provide their subjective judgment about the instrument used for the provided samples. The original questionnaire can be found here[8].

(1) Preserving the musical piece. A different instrument playing the same notes may not always sound subjectively like the same "piece". When this is done in musical contexts, the notes themselves are often changed in order to adapt pieces between instruments, and this is generally referred to as a

---

[6]Link to audio samples for STFT vs. CQT comparison: `https://www.cs.toronto.edu/~huang/TimbreTron/others.html#cqt_stft`

[7]Link to audio samples for generalization: `www.cs.toronto.edu/~huang/TimbreTron/others.html#generalization`

[8]Link to AMT questionnaire for **Does TimbreTron transfer timbre?**: `https://www.cs.toronto.edu/~huang/TimbreTron/AMT_Does_TimbreTron_transfer_Timbre.html`

| **Listen to two audio clips:** (Embedded link for clip A and clip B) |
| :--- |
| The clip A and B may be similar in some ways, and different in others. Rate their similarities with the following criteria: |
| **(i) Instrument similarity:** |
|     (a) Instrument is very similar (e.g. A and B were generated with two different pianos) |
|     (b) Instrument is similar (A and B are in the same family: both wind instrument, or both string instrument, etc) |
|     (c) Instrument is different |
|     (d) I don't know |
| **(ii) Musical piece similarity:** |
|     (a) Musical pieces are nearly identical (e.g. A and B are two different performances of the same piece: perhaps a few notes are different, perhaps timing is slightly different) |
|     (b) Musical pieces are very similar (e.g. A and B are different versions of the same piece, e.g. two different arrangements) |
|     (c) Musical pieces are related (e.g. A and B are two different, but related, pieces) |
|     (d) Entirely different (unrelated) musical piece |
|     (e) I don't know |
| **(iii) What instrument did clip A primarily sound like to you?** |
| **(iv) What instrument did clip B primarily sound like to you?** |

Table 1: The exact question format that was used in the AMT studies. For part (i) and (ii), participants were asked to choose one answer among the options. For part (iii) and (iv), a text box was provided for participants to type in their answers.

| Total Samples | Audio Sample    Answer | Very Similar | Similar | Different | Do not know |
| :--- | :--- | :---: | :---: | :---: | :---: |
| 200 | Target Instrument & TimbreTron Generation | 31.2% | 40.5% | 28.0% | 0.5% |
| 100 | Original Instrument & TimbreTron Generation | 23.0% | 21.0% | 56.0% | 0.0% |

Table 2: AMT results on pair-wise instrument comparisons between our proposed TimbreTron without beam search, ground truth original instrument and ground truth target instrument. This corresponds to question type (i) in Table 1.

new "arrangement" of an existing piece. Thus, even in the cases where we had a recording available in the target domain, the exact notes or timings were not always identical to those in the original recording from which we transferred. Overall, when we did have such a target domain recording of a real instrument, we found that for the pair of (Real Target Instrument, TimbreTron Generated Target Instrument), 67.5% of responses considered the musical pieces to be nearly identical or very similar, while roughly 22.5% considered them related and 10% considered them different. (Details in Table 3.) Thus, it appears that generally the musical piece was indeed preserved.

(2) Transferring the timbre. Evaluating this is challenging because, if the transfer is not perfect (which it is not), then judging similarity of not-quite-identical instruments is fraught with perceptual challenges. With this in mind, we included a range of pairwise comparisons and gave a likert scale with various anchors. Overall, we found that for the pair (Ground Truth Target audio, TimbreTron Generated audio), roughly 71.7% of responses considered the instrument generating the audio to be very similar (e.g. still piano, but a different piano) or similar (e.g. another string instrument). (More details in Table 2.) We also asked participants to identify the instrument that they heard in some of the audio excerpts, with an open-ended question. Generally we found that participants were indeed able to either identify the correct instrument, or confused with a very similar-sounding instrument. For example, one participant described a generated harpsichord as a banjo, which is in fact very close to harpsichord in terms of timbre. As a reference, participants had similar reasonable confusions about identifying ground truth instruments as well (e.g., one participant described a real harpsichord as being a sitar). Based on perceptual evaluations above, we claim that TimbreTron is able to transfer timbre recognizably while preserving the musical content.

| Total Samples | Answer Architecture | Nearly Identical | Very Similar | Related | Entirely Different | Do not know |
|---|---|---|---|---|---|---|
| 200 | Target Instrument & TimbreTron Generation | 29.5% | 38.0% | 22.5% | 10.0% | 0.0% |
| 100 | Original Instrument & TimbreTron Generation | 32.0% | 21.0% | 25.0% | 22.0% | 0.0% |

Table 3: AMT results on pair-wise musical piece comparisons between our proposed TimbreTron without beam search, ground truth original instrument and ground truth target instrument. This corresponds to question type (ii) in Table 1.

| Total Samples | Answer Audio Sample | CQT | same | STFT |
|---|---|---|---|---|
| 400 | STFT+WaveNet counterpart | 54.5% | 23.5% | 22.0% |
| 400 | STFT+Griffinlim counterpart | 55.0% | 25.2% | 19.8% |

Table 4: AMT results on timbre quality comparisons between our proposed TimbreTron, TimbreTron but with STFT Wavenet and TimbreTron with STFT Griffin-Lim. Participants are asked: which one of the following two samples sounds more like the instrument provided in the target instrument sample?

**Comparing CQT vs. STFT**  To empirically test if our proposed TimbreTron with CQT representation is better than its STFT-Wavenet counterpart, or its STFT-GriffinLim counterpart, we conducted a human study using AMT. The original questionnaire can be found here[9] In the questionnaire, we asked Turkers to listen to three audio clips: the original audio from instrument A (the "instrument example"), the TimbreTron generated audio of instrument A, and its STFT conterparts, then asked them: "In your opinion, which one of A and B sounds more like the instrument provided in 'instrument example'"? , where A and B in the questions are the generated samples (presented in random order). Naturally, sounding closer to the "instrument sample" means the timbre quality is better. We conducted two groups of experiment. In the first group, the STFT counterpart is the Wavenet and CycleGAN trained on STFT representation and the result is in first row of the Table 4: most people think the CQT TimbreTron is better. In the second group, we took the same CycleGAN trained on STFT, but instead simply generated the waveform using Griffin-Lim algorithm. The results are in the second row: Even more people think CQT TimbreTron is better. In conclusion, compared to Griffin-Lim as the baseline, training a Wavenet on STFT improved Timbre quality marginally. Furthermore, samples generated by TimbreTron trained on CQT was proven to have significantly better timbre quality.

## 6.5 Ablation Study for TimbreTron

To better understand and justify each modification we made to the original CycleGAN, we conducted an ablation study where we removed one modification at a time for MIDI CQT experiment. (We used MIDI data for ablation because the dataset has paired samples, which provides a convenient ground truth for transfer quality evaluation.) Figure 4 demonstrates the necessity of each modification for the success of TimbreTron.

## 7 Conclusion

We presented the TimbreTron, a pipeline for perfoming high-quality timbre transfer on musical waveforms using CQT-domain style transfer. We perform the timbre transfer in the time-frequency domain, and then reconstruct the inputs using a WaveNet (circumventing the difficulty of phase recovery from an amplitude CQT). The CQT is particularly well suited to convolutional architectures due to its approximate pitch equivariance. The entire pipeline can be trained on unrelated real-world music segments, and intriguingly, the MIDI-trained CycleGAN demonstrated generalization capability to real-world musical signals. Based on an AMT study, we confirmed that TimbreTron recognizably transferred the timbre while otherwise preserving the musical content, for both monophonic and poly-

---

[9]Link to AMT questionnaire for **Comparing CQT vs. STFT**: `https://www.cs.toronto.edu/~huang/TimbreTron/AMT_Comparing_CQT_vs_STFT.html`

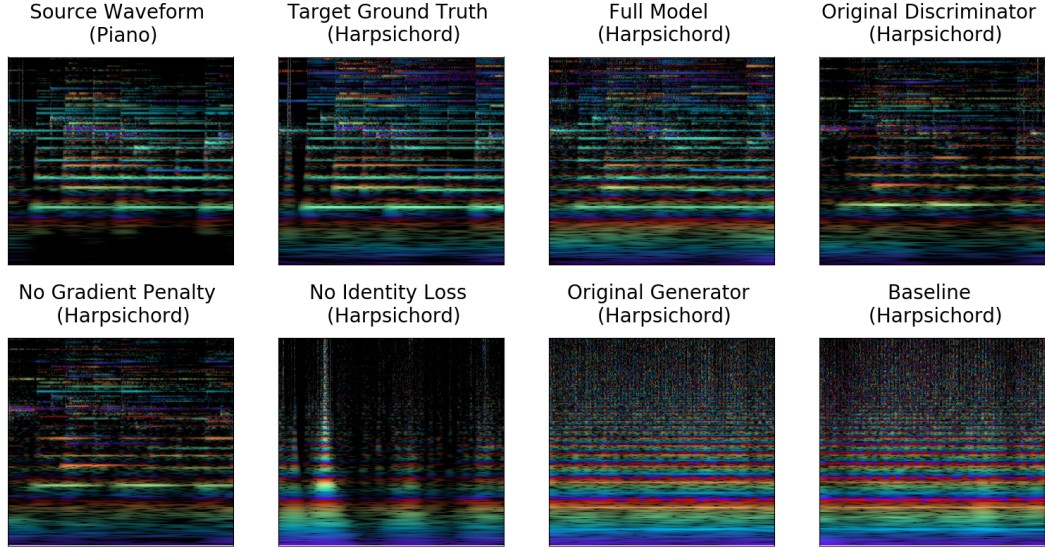

Figure 4: Rainbowgrams of the 4-second audio samples for the ablation study on MIDI test dataset. The source ground truth and the target ground truth come from a paired samples in the dataset. All other audio samples are the timbre transfer results from the source ground truth with different versions (full and ablated) of our TimbreTron. "Full Model" corresponds to the output of our final TimbreTron, which is perceptually closest to target ground truth and have the best audio quality. "Original discriminator" or "Original generator" corresponds to the TimbreTron pipeline with the discriminator or generator replaced by the original discriminator or generator in the original CycleGAN. "No gradient penalty", "No identity loss", and "No data augmentation" refer to the full model without the corresponding modifications. "Baseline" is the original CycleGAN (Zhu et al., 2017)

phonic samples. We believe this work constitutes a proof-of-concept for CQT-domain manipulation of musical signals with high-quality waveform outputs.

## ACKNOWLEDGMENTS

We thank Doug Eck, Jesse Engel, Phillip Isola, Eleni Triantafillou and Sanja Fidler for helpful discussions. We also thank Aidan Gomez and For.ai for early codebase development and coding advice.

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

## A   Components of a Musical Tone

In this section, we will briefly describe the main components of a musical tone: pitch, loudness and timbre (Roederer, 2008).

**Pitch** is described subjectively as the "height" of a musical tone, and is closely tied to the fundamental mode of oscillation of the instrument that is producing the tone. This oscillation mode is often called the *fundamental frequency*, and can often be observed as the lowest band in spectrogram visualizations (Figure 3).

**Loudness** is linked to the perception of sound pressure, and is often subjectively described as the "intensity" of the tone. It roughly correlates with the amplitude of the waveform of the perceived tone, and has a weak dependence to pitch (Hass, 2018).

**Timbre** is the perceptual quality of a musical tone that enables us to distinguish between different instruments and sound sources with the same pitch and loudness (Roederer, 2008). The physical characteristics that define the timbre of a tone are its *energy spectrum* (the magnitude of the corresponding spectrogram) and its *envelope*.

Since sounds generated by physical instruments mostly rely on oscillations of physical material, the energy spectra of instruments consist of bands, which correspond to (approximately) the integer multiples of the fundamental frequency. These multiples are called *harmonics*, or *overtones*, and can be observed in Figure 3. The timbre of an instrument is tightly related to the relative strengths of the harmonics. The spectral signature of an instrument not only depends on the pitch of the tone played, but also changes over time. To see this clearly, consider that a single piano note of duration 500 milliseconds is played in reverse - the resultant sound will not be recognizable as a piano, although it will have the same spectral energy. The *envelope* of a tone corresponds to how the instantaneous amplitude changes over time, and is mainly affected by the instrument's attack time (the transient "noise" created by the instrument when it is first played), decay/sustain (how the amplitude decreases over time, or can be sustained by the player of the instrument) and release (the very end of the tone, following the time the player "releases" the note). All these factors add to the complexity and richness of an instrument's sound, while also making it difficult to model it explicitly.

## B   Spectrogram Processing Details

**Waveform to CQT Spectrogram**   Using constant-Q transform as described in Section 2.1, CQT spectrogram can be easily computed from time-domain waveforms. In this work, we use a 16 ms frame hop (256 time steps under 16kHz), $\omega_0 = 32.70$ Hz (the frequency of C1 [10]), $b = 48$, $k_{max} = 336$ for the CQT transform. Standard implementations of CQT (e.g., librosa (librosa)) also allow scaling the Q values by a constant $\gamma > 0$ to have finer control over time resolution - choosing $\gamma \in (0, 1)$ results in increased time resolution. In our experiments, we choose $\gamma = 0.8$. After the transformation, we take the log magnitude of the CQT spectrogram as the spectrogram representation.

**Waveform to STFT Spectrogram**   All the STFT spectrograms are generated using STFT with $k_{max} = 337$. The window function is picked to be Hann Window with a window length of 672. A 16 ms frame hop is also used (256 time steps under 16kHz). Similar to CQT spectrogram, we also take the log magnitude of the STFT spectrogram as the spectrogram representation after the STFT.

---

[10]C1 refers to the "C1" key, corresponding to the lowest "C" on the piano keyboard.

## C    DETAILED EXPERIMENTAL SETTINGS

### C.1    DATASETS

**MIDI Dataset**    Our MIDI dataset consists of two parts: MIDI-BACH [11] and MIDI-Chopin [12]. MIDI-BACH dataset is synthesized from a collection of bach MIDI files which have a total duration of around 10 hours [13]. Each dataset contains 6 instruments: acoustic grand, violin, electric guitar, flute, and harpsichord. We generated the audio with the same melody but different timbre, which makes it possible to obtain paired data during evaluation.

**Real World Dataset**    Our Real World Dataset comprises of data collected from YouTube videos of people performing solo on different instruments including piano, harpsichord, violin and flute. Each instrument contains around 3 to 10 hours of recording. Here is a complete list of YouTube links from which we collected our Real World Dataset. Note that we've also randomly taken out some segments for the validation set.

- Piano `https://www.youtube.com/watch?v=cOrKeFUZSJ0`
  `https://www.youtube.com/watch?v=GujB0ahKFrY`
  `https://www.youtube.com/watch?v=0sDleZkIK-w&t=629s`

- Harpsichord
  `https://www.youtube.com/watch?v=oeY4a4C-Xuk&t=1555s`
  `https://www.youtube.com/watch?v=Seu9ju7g9u8`

- Violin
  `https://www.youtube.com/watch?v=wtbIT8ALNEA&t=21s`
  `https://www.youtube.com/watch?v=XkZvyA69wCo`

- Flute
  `https://www.youtube.com/watch?v=6GwfuWhOOdY`
  `https://www.youtube.com/watch?v=s6CUi8Gthzc`
  `https://www.youtube.com/watch?v=uE9SjAqPGsc&t=1001s`

### C.2    DOMAIN SPECIFIC GLOBAL NORMALIZATION

As is shown in Figure 5 in Appendix C.7, the distribution of spectrogram pixel magnitude is roughly centered at -2, which is not good for learning because of the tanh activation function works better when the activation is in the range of $[-1, 1]$. Thus, we globally normalized the spectrogram data to be mostly in the range of $[-1, 1]$ for each instrument domain. We scaled and shifted the spectrograms based on the mean and standard deviation of each instrument domain to achieve Domain Specific Global Normalization in the input pipeline, and reverse this operation on the output of CycleGAN to minimize possible distribution shift before feeding the output for wavenet generation.

### C.3    CYCLEGAN TRAINING DETAILS

In CycleGAN training, because we made several architectural changes, we retuned the hyperparameters. The weighting for our cycle consistency loss is 10 and the weighting of the identity loss is 5. In the original CycleGAN the weighting of identity loss is constant throughout training but in our experiment, it stays constant for the first 100000 steps, then it starts linearly decay to 0. We set the weighing for Gradient Penalty to be 10, as was suggested in Gulrajani et al. (2017). Our learning rate is exponentially warmed up to 0.0001 over 2500 steps, stays constant, then at step 100000 starts to linearly decay to zero. The total training step is 1.5 million steps, trained with Adam optimizer (Kingma and Ba, 2014) with $\beta_1 = 0$ and $\beta_2 = 0.9$, with a batch size of 1.

---

[11]from website `www.jsbach.net/midi/`
[12]from website `www.piano-midi.de/chopin.htm`
[13]For all the synthesized audio, we use Timidity++ synthesizer

### C.4 Conditional Wavenet Training

For the conditional wavenet , we used kernel size of 3 for all the dilated convolution layers and the initial causal convolution. The residual connections and the skip connections all have width of 256 for all the residual blocks. The initial causal convolution maps from a channel size of 1 to 256. The dilated convolutions map from a channel size of 256 to 512 before going through the gated activation unit. The conditional wavenet is trained with a learning rate of $0.0001$ using Adam optimizer (Kingma and Ba, 2014), batch size of 4, sample length of 8196 ($\approx 0.5s$ for audio with $16000Hz$ sampling rate). To improve the generation quality we maintain an exponential moving average of the weights of the network with a decaying factor of $0.999$. The averaged weights are then used to perform the autoregressive generation. To make the model more robust, we augmented the training dataset by randomly rescaling the original waveform based on its peak value based on a uniform distribution $uniform(0.1, 1.0)$. In addition, we also added a constant shift to the spectrogram before feeding it into the WaveNet as the local conditioning signal; this shift of +2 was chosen to achieve a mean of approximately zero.

### C.5 Beam Search

During autoregressive generation, we perform a modified beam search where the global objective is to minimize the discrepancy between the target CQT spectrogram and the CQT spectrogram of the synthesized audio waveform. Our beam search alternates between two steps: 1) run the autoregressive WaveNet on each existing candidate waveforms for $n$ steps ($n = 2048$) to extend the candidate waveforms, 2) prune the waveforms that have large squared error between the waveforms' CQT spectrogram and the target CQT spectrogram (beam search heuristic). We maintain a constant number of candidates (beam width = 8) by replicating the remaining candidate waveforms after each pruning process. To make sure the local beam search heuristic is approximately aligned with the global objective, we take $n$ extra prediction steps forward and use the extra $n$ samples along with the candidate waveforms to obtain a better prediction of the spectrogram for the candidate waveforms. The algorithm is provided in details as follows given the target spectrogram $C_{target}$:

1. $k \leftarrow 0$
2. Perform $2n$ autoregressive synthesis step on WaveNet on $\{x_1, \cdots, x_k\}$ with $m$ parallel probes ($m$ is the beam width) to produce $m$ subsequent waveforms: $\{x_{k+1}^{(1)}, \cdots, x_{k+2n}^{(1)}\}, \{x_{k+1}^{(2)}, \cdots, x_{k+2n}^{(2)}\}, \cdots, \{x_{k+1}^{(m)}, \cdots, x_{k+2n}^{(m)}\}$.
3. Compute the CQT spectrogram $C_i$ of $\{x_{k+1}^{(i)}, \cdots, x_{k+2n}^{(i)}\}$ for each $i \in \{1, 2, \cdots, m\}$, and find the waveform $\{x_{k+1}^{(i')}, \cdots, x_{k+2n}^{(i')}\}$ with the lowest square difference between $C_i$ and the target CQT spectrogram $C_{target}$
4. Update the waveform $x_j = x_j^{i'}, \forall j \in \{k+1, k+2, \cdots, k+n\}$
5. $k \leftarrow k + n$

### C.6 One-shot generation of longer segments

In our earlier attempts, we tried generating 4 seconds segments and then merge them back. However, this resulted in volume inconsistencies between the 4 second generations. We suspect the CycleGAN learned a random volume permutation, because essentially there's no explicit gradient signal against it from the discriminator, after we enabled volume augmentation during train time. To resolve this issue, we removed the size constraint in our generator during test time so that it can generate based on input of arbitrary length. At test time, the dataset is no longer 4 second chunks, instead, we preserved the original length of the musical piece(except when the piece is too long we cut it down to 2 minutes due to GPU memory constraint). During test time generation, the entire piece is fed into the CycleGAN generator in one shot.

### C.7 Spectrogram raw pixel intensity histogram

Figure 5 shows that the rough distribution of spectrograms are centered at -2. As is discussed in Section 3.1, we globally normalized our input data based oh the distribution of spectrograms for each domain of instruments.

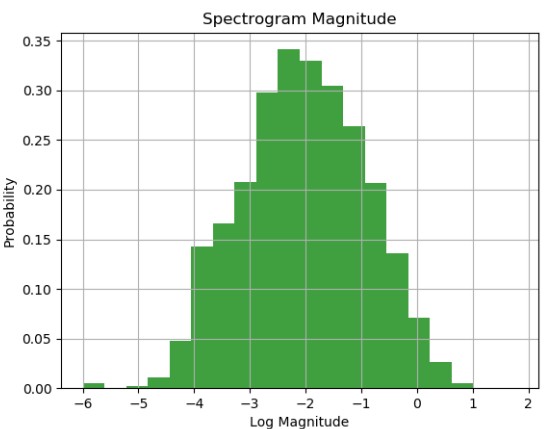

Figure 5: Spectrogram raw pixel intensity histogram

