# OpenReview forum: "TimbreTron: A WaveNet(CycleGAN(CQT(Audio))) Pipeline for Musical Timbre Transfer"
_ICLR.cc/2019/Conference_

### Official Review · AnonReviewer1 · 2018-11-02
**Compelling results on timbre transfer, backed up by human evaluation**

**Rating:** 8
**Confidence:** 4

**Review:**

Summary
-------
This paper describes a model for musical timbre transfer.
The proposed method uses constant-Q transform magnitudes as the input representation, transfers between domains (timbres) by a CycleGAN-like architecture, and resynthesizes the generated CQT representation by a modified WaveNET-like decoder. The system is evaluated by human (mechanical turk) listening studies, and the results indicate that the proposed system is effective for pitch and tempo transfer, as well as timbre adaptation.


High-level comments
-------------------

This paper is extremely well written, and the authors clearly have a great attention to detail in both the audio processing and machine learning domains.  Each of the modifications to prior work was well motivated, and the ablation study at the end, while briefly presented, provides a good sense of the contributions of each piece.

I was unable to listen to the examples provided by the link in section 6, which requires a Microsoft OneDrive login to access.  However, the youtube link provided in the ICLR comments gave a reasonable sample of the results of the system.  Overall, the outputs sound compelling, and match my expectations given the reported results of the listening studies.

On the quantitative side, it would have been nice to see a measurement of phase retrieval of the decoder component, which could be done in isolation from the transfer components by feeding in original CQT magnitudes.  This might help give a sense of how well the model can be expected to perform, particular as it breaks down along target timbres.  I would expect some timbres to be easier to model than others, and having a quantitative handle on that could help put the listener study in a bit more perspective.

Detailed comments
-----------------

The paper contains numerous typos and grammatical quirks, e.g.:
    - page 5: "GP can stable GAN training"
    - page 7: "CQT is equivalent to pitch"


The reverse-generation trick in section 3.2 was clever!

---

> ### Author Response · Authors · 2018-11-19
> **Response to R1. Thank you for your review, which allowed us to improve our work**
>
> Thanks for catching the typos! They are now fixed. GP means Gradient Penalty; sorry for not writing down the abbreviation in the first place, it is now fixed.
>
> As for quantitative measurement of phase retrieval of the decoder component, we will include qualitative comparison of rainbowgram(which encodes phase information with color) of the source audio and the wavenet reconstruction (wavenet(CQT(source audio)) and you can find the samples here:https://1drv.ms/f/s!ApC93lRyk9iagZ5NBNV_ERqOJFK_3w. The reason why we don’t think in our particular case that quantitative measurement can provide much insight in this regard is because phase retrieval is neither the focus nor a sub-goal of our work: so if, for example, our system produces an output that's shifted by some number of time steps, it would be a perfectly good output, even though the phases are all completely wrong.
>
> We apologize for the complexity of listening to samples via OneDrive… In testing it anonymously in advance, we did not have any of these errors, so we did not anticipate such a problem. We are glad you were able to access the youtube link, and we are in the progress of making a project page for our camera ready version, and by then it should be easier to go through all the samples.

---

### Official Review · AnonReviewer3 · 2018-11-03
**Timbre can be tranferred pretty well using a constant-Q transform for features, followed by a CycleGAN to do the transfer, followed by a Wavenet to resynthesize it to audio.**

**Rating:** 7
**Confidence:** 4

**Review:**

Main Idea: The authors use multiple techniques/tools to enable neural timbre transfer (converting music from one instrument to another, ex: violin to flute) without paired training examples. The authors are inspired by the success of CycleGANs for image style transfer, and by the success of Wavenet for generating realistic audio waveforms. Even without the CycleGAN, the use of CQT->WaveNet for time stretching and pitch shifting of a single piece is an interesting and valuable contribution.

Methodology: Figure 1 captures the overall timbre-conversion methodology concisely. In general the details of the methodology look sound. The lengthy appendices offer additional implementation details, but without access to a source code repository, it is hard to say if the results are perfectly reproducible.

Experiment and Results: Measuring the quality of generated audio is challenging.  To do so, subjective listening tests are conducted on Amazon mechanical turk, but without a comparison to a baseline system except for another performance of the target piece. Note that there are few published timbre-transfer methods (see Similar Work).

One issue with the AMT survey is that the total number of workers is not reported, and as such the significance of the results can be questioned.

Significance: In my mind, the paper offers validation of the three techniques used. CycleGANs, originally designed for images,  are shown to work for style transfers on audio spectrograms. Wavenet's claim to be a generic technique for audio generation is tested and validated for this domain (CQT spectrogram to audio). That CQT outperforms STFT on musical data seems to be a well established result already, but this offers further validation.

This paper also offers practical advice for adapting the techniques/tools (Wavenet, CycleGAN, CQT) to the timbre-transfer task.


Similar Work:

I have only found 2 papers dedicated to timbre transfer in the field of Learning Representations.

Bitton, Adrien, Philippe Esling, and Axel Chemla-Romeu-Santos. "Modulated Variational auto-Encoders for many-to-many musical timbre transfer." arXiv preprint arXiv:1810.00222 (2018).

which was published on sept 29th 2018, so less than 30 days ago, which is fine according to the reviewer guidelines.


Verma, Prateek, and Julius O. Smith. "Neural style transfer for audio spectograms." arXiv preprint arXiv:1801.01589 (2018).

which is a short 2 page exploratory paper.


It could be useful to cite:

Shuqi Dai, Zheng Zhang, Gus G. Xia.  "Music Style Transfer: A Position Paper." arXiv preprint arXiv:1803.06841 (2018)


Writing Quality

Overall the paper is written well with clear sentences.

Certain key information would be useful to move from the appendices to the main body of the paper.  This includes the number of AMT workers, the size of the CQT frame/hop over which they are summarized, and the set of instruments that are being used in the experiments.


Some minor nitpicks:

section 6.3, sentence 2 needs to be reworked. ('After moving on to real world data, we noticed that real world data is harder to learn because compared to MIDI data it’s more irregular and more noisy, thus makes it a more challenging task.')

section 3.2 sub-section 'Reverse Generation', sentence 1 uses the word 'attacks' for the first time. Please explain this for those not familiar.

section 3.1, sentence 3 has a typo, 'Thanks' is wrongly capitalized.

table 1 (and other tables in appendix), 'Percentage' (top left) does not add anything to the table.

---

> ### Author Response · Authors · 2018-11-19
> **Response to R3. Thank you for your review, which allowed us to improve our work**
>
> We would like to thank Reviewer3 for the helpful and high quality review! It has enabled us to add some missing information to our paper and improve the writing quality.
>
> As for the repo, we are currently working on cleaning up the code and will release once we finish. As for the baseline comparison of our work, we did include a detailed ablation study comparing our final model with various baseline and “partial” models(subtracting a modification each time)
>
> Thanks for bringing up three papers that are related to ours. We indeed cited work by Verma. et al in our original paper. We now cite work done by Dai. et al and Bitton. et al in the updated paper. Thanks!
>
> As for the detailed information on the number of AMT workers (30 workers per questionnaire, but number of questions per questionnaire varies, for example, we asked 8 question per worker for comparing STFT and CQT experiment, thus in total we have 240 data points. See more details in the paper), the size of the CQT frame/hop over which they are summarized(16ms frame hop (256 time steps under 16kHz)), and the set of instruments(Piano, Violin, Flute, Harpsichord) that are being used in the experiments, they are updated in the main body of the paper.
>
> Section 6.3, sentence 2 was reworked.
>
> Attack is now explained in section 3.2 as the onset characteristics of an instrument in which it reaches a large amplitude quickly.
>
> Thanks for catching the “thanks” typo. It’s now fixed.
>
> Thanks for bringing up the percentage being meaningless, it’s now removed.
>
> Again, thanks for your very constructive comments which allowed us to improve our paper!

---

> > ### Public Comment · (anonymous) · 2020-04-17
> > **When will you release the code?**
> >
> > > As for the repo, we are currently working on cleaning up the code and will release once we finish.
> >
> > Though it's been almost a year and a half since you wrote in the rebuttal, the source code has still not uploaded yet in the repository specified in the paper.
> > Considering that the content of the rebuttal can be a factor to be accepted, I think you must be responsible for what you wrote in your rebuttal as a member of the scientific community.

---

### Official Review · AnonReviewer2 · 2018-11-05
**interesting idea, but weak experimental validation, and too much of combining black-boxes**

**Rating:** 4
**Confidence:** 5

**Review:**

The paper proposes a method for converting recordings of a specific musical instrument to another. The proposed approach is apply CycleGAN, which was developed for image style transfer, to transfer spectrograms. The synthesis is done using WaveNet.

The paper is interesting in the core idea. It demonstrates that this combination of building blocks can indeed map recordings while achieving certain characteristics of the target instrument.

The paper correctly describes "timbre" as a catch-all term for characterizing instruments besides pitch and volume. The success of the method should be judged along two dimensions, which are both very subjective:
 - Does the method transfer "enough" of the target instrument's characteristics?
 - Is the resulting audio quality sufficient?

The paper is easy to follow for someone with background in signal processing. I believe it is sufficiently easy to follow for readers with general computer-science and machine-learning background.

The paper focusses a lot on the choice of spectral representation. It compares short-term Fourier transform (STFT), which is the generic standard, and Constant-Q Transforms (CQT), a variant of STFT that uses a logarithmic frequency axis. To someone with signal-processing background, the choice of CQT seems logical and not something that would be challenged strongly as long as a simple comparison to STFT confirms that it works a bit better. I find the comparison between the two too dominant in the paper, and distracting from the other issues that I feel are more important (see below). For example, Section 6.2 states "We demonstrate that the aforementioned translation does indeed result in a perceivable pitch shift when fed to our conditional WaveNet." But that is trivial: Since changing the playback sample rate by a few half steps does not fundamentally alter the perceived timbre of an instrument, and such a change will, by construction of CQT, shift the CQT representation, and since the WaveNet has seen examples of the source instrument for all notes, shifting the CQT respresentation must necessarily result in a perceived pitch change in the re-synthesized wveform that does not fundamentally change its timbre. The real question here is whether the reconstructed signal is of the same quality as, for example, a simple PSOLA-based pitch change would be.

A larger problem of the paper is that the result section seems to only test two instrument mappings, violin to flute, and piano to harpsichord. One notes that these instrument pairs mostly differ in spectral envelope, while they are rather similar in longer-term temporal variations, such as what is sometimes characterized as the ADSR curve (attack-decay-sustain-release) and vibrato. These, in my view, are very important aspects of a musical instrument's characteristics, which are not addressed by the paper. (Whether they are considered part of "timbre" is not clear, but without mapping these, one cannot meaningfully speak of mapping instruments, which is the end goal of this paper.)

Another big problem in my view is that the audio quality is just not good. I hear a lot of musical-noise artifacts and local timbre modulations. Also it is not clear why the source material is of poor quality (sounds quite noisy, most likely in part due to mu-law 8-bit encoding, and they sound like 11 kHz recordings), for which there is no justification in 2018.

Lastly, I am not happy with the "beam-search" approach. That approach is used to post-correct imperfections in the WaveNet synthesis. It samples multiple generation hypotheses, and re-weights hypotheses by how well they match the original CQT when converted back. The need for this indicates a fundamental flaw in the WaveNet synthesizer. The authors explicitly say they did not want to fix the WaveNet algorithm itself. In my view, this is what should have been done.

The authors should focus much more on how to achieve sufficient WaveNet synthesis quality. This should be the main bulk of the paper, and would be a requirement for me to accept the paper.

So overall, the paper feels a little too much of combining black boxes.

In terms of significance, I would not think that this paper is getting near solving this problem, hence I rate it of less significance in the current state of results.

Pros:
 - interesting idea
 - reasonable approach by combining existing building blocks

Cons:
 - too much focus on STFT vs. CQT
 - too little focus on getting WaveNet synthesis right
 - too limited experimental validation (too restricted choice of instruments)
 - poor resulting audio quality
 - feels too much of combining black boxes

As a result I rate the paper "not good enough" in its current form.

---

> ### Author Response · Authors · 2018-11-19
> **Response to R2. Thank you for your review, which allowed us to improve our work**
>
> We would like to thank Reviewer2 for the insightful, high quality and clear review! It has allowed us to greatly improve our work and our paper. We hope the revised draft of the paper with the clarifications and improvements made below serve to increase your rating of our work.
>
> 1.	In choosing a CQT rather than STFT representation, we agree that, devoid of any other context, such a choice might indeed not be contentious, especially from a signal processing perspective. But the context here is that we are not just using this for discrimination or analysis purposes, but optimizing reconstruction is also essential: since Griffin-Lim is not possible for CQT, then in this case, switching from STFT to CQT means that we need to fundamentally change the pipeline. Furthermore, STFT and Grifflin-Lim is still widely used in state of the art speech and music research, for example in Tacotron 2 by Shen et al.(2018) and Deep Voice 3 by Ping et al.(2018).  Hence, we needed to demonstrate that the improvement from CQT in this context is significant enough to justify the added complexity of the WaveNet synthesizer.
>
> 2.	In advance of our experiments, it would certainly be a reasonable conjecture that the spectrogram-shifting trick would work for pitch shifting. But we don’t think this was so “obvious” as to not require experimental confirmation.
>
> 3.	Thank you for suggesting the comparison of our pitch shifted samples with PSOLA-based pitch change. We did some PSOLA experiments for baseline to address your concern. The results can be found here: https://1drv.ms/f/s!ApC93lRyk9iagZ5pKpRLdBcdJ1xJbw As you can see, PSOLA performs well on simple samples, like notes_shifted.wav However, it completely fails on more complicated monophonic samples. We think that’s due to the fact that for PSOLA to work well, fundamental frequency needs to be precisely detected, which itself is not a trivial task for a complex monophonic musical piece. We haven’t put this in the paper yet because we are not sure this is a proper baseline and we are still working on better baseline comparison, and will add the comparison on to paper later.
>
> 4.	We did in fact run more than two instrument pairs, and we encourage you to take a look at the example outputs we provided in the OneDrive folder, as well as our final demo video(https://www.youtube.com/watch?v=aT4D4mTITko). In the OneDrive folder Section 7 (which corresponds to the Section 7. Conclusion of the paper), we also included Violin to Flute and Piano to Violin in our original submission. (The main obstacle to reporting more instrument pairs was simply the computational cost of training additional CycleGANs; our computational resources were much more limited than some prominent groups publishing on deep learning for music.) The idea of this work is not to develop a commercial level product for artists, but rather we are only aiming at empirically verifying that our methodology is a practical approach to the problem of Timbre Transfer, a proof of concept. But we agree that the more instrument pairs the better, thus in this folder: https://1drv.ms/f/s!ApC93lRyk9iagZ5M91jko9nSAiYMiA, we also included some samples of Violin -> Piano(Sample 19) and Harpsichord -> Piano(Samples 16 and 18) and Piano -> Flute(sample 17). The latter is a particularly interesting failure case where TimbreTron dreamed up all the “piano vibrato” from the long notes in the source flute sample.  We will gradually add more instruments and translation directions, including both successes and other failure cases as well.
>
> 5.	Thank you for bringing up ADSR and vibrato. These are indeed interesting cases where it’s ambiguous what the “correct” mapping is. Because TimbreTron does not simulate the physics of the instruments, it sometimes transfers effects that would be impossible or unusual for the target instrument. We highlighted two such examples in our demo video: a crescendo on a sustained piano note, and a string ensemble pausing to breathe. In the context of TimbreTron, we consider these to be interesting artifacts. If one wants to build a commercial system to produce convincing instrument samples, one would want to somehow remove these artifacts. (On the other hand, they might be beneficial, insofar as they enable a broader range of expression in the target instrument, as was the case for our generated harpsichord for the Moonlight Sonata.)
>
> (due to characters limit, more response will follow in the next post)
> ...

---

> > ### Author Response · Authors · 2018-11-19
> > **Continued Response to R2**
> >
> >
> > ...
> >
> > 6.	For the poor audio quality in the source material, could you point us to some specific samples? We used 16kHz as our raw source audio and we chose that because even in 2018, the significant majority of audio-related ML research is done with 16 kHz audio, (For example, in recent research on audio superresolution, https://kuleshov.github.io/audio-super-res/ , the authors describe their 16kHz samples as their high-quality dataset) and perceptually it was clear enough to tell details of timbre. Furthermore, the focus of this research is to get timbre transferred correctly while preserving other musical content. Audio quality is part of musical content and is not within the problem we’re trying to solve. For example, we are not trying to do super resolution on audio, and I think it’s fair to say 16 kHz is the standard resolution, corresponding to the 256x256 resolution used in preprocessed ImageNet in CV research. So far we haven’t found any sample that’s 11kHz as you suggested, but we’ll make sure all source audio in the updated paper will be the standard 16kHz resolution.
> >
> > 7.	Due to potential copyright issue, we didn’t find enough samples in other instrument online that we are absolutely certain won’t have any copyright issue, so we recorded our own samples, thus didn’t have a lot of them.
> >
> > 8.	As for the beam search, we are aware that beam search should not be a key component on which the entire model relies in order to work (and if that were the case, then we agree that that would be problematic). However, the system does not rely on it: the improvement is only subtle and our results are still acceptable without it. As shown in the original paper, the AMT results we’ve shown are actually done without beam search. We only included it in the paper because it can still marginally improve the generated audio quality. We added some samples generated by TimbreTron without any beam search and they can be found here: https://1drv.ms/f/s!ApC93lRyk9iagZ5M91jko9nSAiYMiA
> >
> > 9.	Due to the commercial success of WaveNet, well-resourced company labs are already working hard on improving it, and we look forward to being able to integrate such improvements into our pipeline. Hence, improving WaveNet itself wasn’t a major focus of our work. Instead, we gave two simple methods which adapted WaveNet to our task and pipeline -- beam search and reverse generation -- and we expect these tricks would apply equally well to future improved WaveNet-like models. We added some samples that were reconstructed by WaveNet from source audio with and without beam search and they can be found here: https://1drv.ms/f/s!ApC93lRyk9iagZ5NBNV_ERqOJFK_3w.  We are in progress of adding more detailed ablation study on the wavenet component that, hopefully, will show the impact of reverse generation, data augmentation and beam search.
> >
> > 10.	We are in progress of making a project page so that all audio samples can be more organized in the camera ready version.
> >
> > Again, we sincerely appreciate such a detailed and insightful critique of our work. Please let us know of any other changes that would improve our work. Thanks!

---

> > > ### Author Response · Authors · 2018-11-26
> > > **WaveNet ablation study added**
> > >
> > > We added in Appendix E the ablation study focusing on the WaveNet component of TimbreTron. It showed the impact of each of our modification (reverse generation, data augmentation and beam search) on the output quality of WaveNet. The corresponding audio samples can be found here: https://1drv.ms/f/s!ApC93lRyk9iagZ8qP0IlxLXZkbO-iA

---

> > > > ### Comment · AnonReviewer2 · 2018-12-13
> > > > **the new examples confirm to me that the approach is not ready**
> > > >
> > > > Thanks for the detailed response and extra effort to address some of my points. However, listening to the new examples, I have come to an even stronger opinion that the proposed method is not ready. From the samples, it is clear that WaveNet and the mapping method both introduce artefacts that cause unacceptable loss of quality. Yes, when mapping flute to piano it definitely does have characteristics of piano, but at the cost of extreme distortions that are far away from being useful for any purpose at this point in time.
> > > >
> > > > I do believe that this is an interesting idea and an interesting line of research, but I also continue to believe that the results are not good enough at this point in time.
> > > >
> > > > Some details:
> > > >
> > > > > "We demonstrate that the aforementioned translation does indeed result in a perceivable pitch shift when fed to our conditional WaveNet." I believe my original comment is still correct, "The real question here is whether the reconstructed signal is of the same quality as, for example, a simple PSOLA-based pitch change would be."
> > > >
> > > > I believe that a research system should be better than commercially available systems, or at least be on par. Please check out this Youtube video that uses Audacity to time-stretch an audio clip: https://www.youtube.com/watch?v=SjVY2Fs8-24. Although this is time-stretching and not pitch-changing, obviously pitch-shifting can be implemented by time-stretching followed by playing back at a different sampling rate. Their example time-stretches a pop song by 30%. Please play back from 3:50. With concentration, I can hear some minor artefacts, but by and large, it sounds good and not unpleasant at all.
> > > >
> > > > Compare this to the pitch change examples "Mozart up/down3.wav", which has strong distortions which are very unpleasant. Using WaveNet to reconstruct *originals* seems to work, by and large, except for quite some additional noise, but even the basic task of pitch-shifting already shows strong artefacts.
> > > >
> > > > This shows me that WaveNet synthesis itself is inadequate in its current form.
> > > >
> > > > > "We did some PSOLA experiments for baseline to address your concern"
> > > >
> > > > The shifted_bumble.wav example in https://onedrive.live.com/?authkey=%21ACqUS3QXHSdcSW8&id=9AD8937254DEBD90%2120331&cid=9AD8937254DEBD90
> > > >
> > > > is full of crackling artefacts. I have never heard such artefacts for any re-pitching algorithm. I think you did not do the windowing right.
> > > >
> > > > > "as well as our final demo video(https://www.youtube.com/watch?v=aT4D4mTITko)"
> > > >
> > > > Youtube says "This video is unavailable." Did Youtube kill it due to copyright claims? Then please post the video on the OneDrive as well.
> > > >
> > > > > "we also included some samples of Violin -> Piano(Sample 19) and Harpsichord -> Piano(Samples 16 and 18) and Piano -> Flute(sample 17)"
> > > >
> > > > This example shows strong artefacts in that it synthesizers frequencies below the original pitch. It sounds like a base guitar playing unisono with the main melody (and sometimes a different note). Does this come from WaveNet or the mapping method? Either way, my takeaway is, again, that the method is not ready.
> > > >
> > > > > "For the poor audio quality in the source material, could you point us to some specific samples?"
> > > >
> > > > This is an example which is very noisy, has some strange volume dip at the very beginning that is impossible to produce with a piano, and a drop-out at 3.5 seconds: https://onedrive.live.com/?authkey=%21ACqUS3QXHSdcSW8&id=9AD8937254DEBD90%2120329&cid=9AD8937254DEBD90
> > > >
> > > > It is, however, possible that the poor quality I perceived is an byproduct of OneDrive's audio-playback interface, which may send compressed audio when playing uncompressed WAV files. So I concede this point.

---

> > > > > ### Author Response · Authors · 2018-12-14
> > > > > **Response to R2. Thanks for the comments!**
> > > > >
> > > > > Thanks for your comments!
> > > > >
> > > > > - Sorry about the broken YouTube link, that link stopped working and please use this link instead: https://youtu.be/2ypcAZRYZJg  We checked that this one is working.
> > > > >
> > > > > - We agree with your point that, at least in certain contexts, a research system should be better than commercially available systems, or at least be on par. However our work is about Timbre transfer, which doesn’t have an existing commercially available system yet. The pitch shift and time stretch experiment were just included to demonstrate that we can get those side benefits for free by training TimbreTron on CQT representation; it was not our primary goal, and was not allocated the majority of space in the paper as well. We agree that if the sole purpose of our system had been to do pitch shifting or time stretching for music, then existing commercial tools would have been an important baseline to consider.
> > > > >
> > > > > - As for the audio quality, one way we chose to address the subjective nature of the question was by conducting AMT human study. Based on our human study results there was strong evidence that TimbreTron is indeed able to transfer Timbre recognizably while preserving other musical content.

---

### Public Comment · ~Keunwoo_Choi1 · 2018-10-06
**Some details of the system are missing**

Thanks for the work. I'd expect more details of the system. For example, what's the kernel sizes and number of channels of the WaveNet? With only the number of layers and their dilation rates it's not possible to understand/reproduce the proposed system. Details of the discriminator should be provided as well.

I'd also appreciate more details of the datasets, e.g. who are the composers, what kind of music it is (more than simply 'classical music'), etc. One important information would be if they are polyphonic.

Finally, the OneDrive link does not work at the moment.

---

> ### Author Response · Authors · 2018-10-09
> **Thanks for the comment!**
>
> Hi Keunwoo, thanks for the comment!
>
> For WaveNet, we used kernel size of 3 for all the dilated convolution layers and the initial causal convolution. The residual connections and the skip connections all have width of 256 for all the residual blocks. The initial causal convolution maps from a channel size of 1 to 256. The dilated convolutions map from a channel size of 256 to 512 before going through the gated activation unit. In addition, we also added a constant shift to the spectrogram before feeding it into the WaveNet as the local conditioning signal; this shift of +2 was chosen to achieve a mean of approximately zero. These details will be updated in the paper.
>
> In terms of the discriminator, we adopted the original discriminator architecture from the original CycleGAN paper except that we ran it on the full signal rather than random patches, as was discussed in section 4.
>
> The dataset we used for training both the CycleGAN and the wavenet consists of real world recordings that contain only a single timbre, collected from YouTube. We have not filtered the samples based on whether they are polyphonic or not - for instruments that support polyphony (such as piano and harpsichord), the majority of the recordings are polyphonic. You can see the full list of the recordings we used in our experiments in Appendix C.1 in the updated paper.
> As for the OneDrive link, there seems to be an issue with the OpenReview redirection. However if you copy the link and paste it in your browser then it should work.

---

### Author Response · Authors · 2018-11-02
**Demo Video**

We made a demo video for this work, and we strongly encourage you to watch it as it will give you a general and intuitive idea about this work. You can find it here:  https://youtu.be/2ypcAZRYZJg

---

### Author Response · Authors · 2018-11-19
**General updates to reviewers**


1.	We included more samples that were generated by TimbreTron but without any beam search here: https://1drv.ms/f/s!ApC93lRyk9iagZ5M91jko9nSAiYMiA

2.	We included samples reconstructed by WaveNet that were not transferred by CycleGAN in order to show that without phase information, our WaveNet can reconstruct waveform from CQT pretty well (i.e, WaveNet(CQT(source audio))). The samples can be found here:https://1drv.ms/f/s!ApC93lRyk9iagZ5NBNV_ERqOJFK_3w

3.	We are in progress of making a project page so that all audio samples will be more organized in the camera-ready version.

4.	We added the original user interface of our AMT experiments and they can be found here:https://1drv.ms/f/s!ApC93lRyk9iagZ0ndQIYdJBAqYDlDA

5.     We added in Appendix E the ablation study focusing on the WaveNet component of TimbreTron. It showed the impact of each of our modification (reverse generation, data augmentation and beam search) on the output quality of WaveNet. The corresponding audio samples can be found here: https://1drv.ms/f/s!ApC93lRyk9iagZ8qP0IlxLXZkbO-iA

---

### Public Comment · ~Rahul_Bhalley1 · 2018-12-29
**Incomplete information about one-shot generation of long musical pieces**

Congratulations for good research work! But it misses some technical details.

Firstly in appendix C.6, there is no information given about how the size constraint in generator network was removed for processing arbitrary length inputs. Preserving the musical length to at most 2 minutes (due to GPU memory constraint, as written in paper) will be a long 7680 x 256 image whereas the generator is designed to process 256 x 256 image inputs (in accordance to original CycleGAN paper details). The explicit details about how the 7680 x 256 image is fed to the generator in one-shot must be provided.

Secondly there is no discussion given about setting up the batch size equal to 1 i.e.:
1. Is it due to limited computation resources like CycleGAN research https://github.com/junyanz/pytorch-CycleGAN-and-pix2pix/issues/198
2. Does higher batches not result in better quality of timbre style transfer?
3. Is it just a matter of choice?

---

> ### Author Response · Authors · 2018-12-30
> **Thanks for your interest and the comment!**
>
> Hi Rahul,
>
> Thanks for your interest in our work and thanks for the comment!
>
> Theoretically there is no size constraint in the generator network as it’s fully convolutional. However practically because the generator was initially written in TensorFlow, we initially specified the input placeholder with size [1, 257, 251] (Note that it’s no longer 256 as in the image case) and that was the “size constraint”. We removed it (so that it can take in arbitrary size during test time) to address the volume jump issue. This will be further clarified in the appendix C.6 of the camera-ready version.
>
> And yes it was due to limited computation resources. We did not experiment with larger batch size.
>
> Please let us know if you have any further question, thanks!

---

### Meta-Review · Area_Chair1 · 2018-12-13
**Interesting paper, has some issues, but would be of interest to the community**

**Confidence:** 4
**Recommendation:** Accept (Poster)

**Metareview:**

Strengths: This paper is "thorough and well written", exploring the timbre transfer problem in a novel way. There is a video accompanying the work and some reviewers assessed the quality of the results as being good relative to other approaches. Two of the reviewers were quite positive about the work.

Weaknesses: Reviewer 2 (the lowest scoring reviewer) felt that the paper was a little too far from solving the problem to be of high significance and that there was:
 - too much focus on STFT vs. CQT
 - too little focus on getting WaveNet synthesis right
 - too limited experimental validation (too restricted choice of instruments)
 - poor resulting audio quality
 - feels too much of combining black boxes

AMT listening tests were performed, but better baselines could have been used.
The author response addressed some of these points.

Contention:
An anonymous commenter noted that the revised manuscript added some names in the acknowledgements, thereby violating double blind review guidelines. However, the aggregated initial scores for this work were past the threshold for acceptance. Reviewer 2 was the most critical of the work but did not engage in dialog or comment on the author response.

Consensus:
The two positive reviewers felt that this work is worth of presentation at ICLR. The AC recommends accept as poster unless the PC feel the issue of names in the Acknowledgements in an updated draft is too serious of an issue.